# Information and communication technologies and quality of life in home confinement: Development and validation of the TICO scale

José Antonio García del Castillo-Rodríguez[1☯], Irene Ramos-Soler[2☯], Carmen López-Sánchez[2☯], Carmen Quiles-Soler[2☯]*

1 Department of Health Psychology, Miguel Hernández University, Elche, Alicante, España, 2 Department of Communication and Social Psychology, University of Alicante, San Vicente del Raspeig, Alicante, España

☯ These authors contributed equally to this work.
* mc.quiles@ua.es

## Abstract

The mandatory home confinement of the Spanish population, implemented in response to the COVID-19 pandemic, presents a unique opportunity to study the use and influence of Information and Communication Technologies (ICT) in people's perception of quality of life during this exceptional situation. This article adapts and validates a psychometric scale designed to identify and measure the main dimensions of the Quality of Life construct perceived through ICT use. To this end, an exploratory and transversal study has been carried out in Spain on a sample of 2,346 participants. Data processing has been carried out with SPSS and EQS. The results provide evidence of the reliability and psychometric quality on the scale, which exhibits adequate consistency that facilitates its application. The confirmatory factor analysis showed a hierarchical model of three correlated factors that account for the dimensions "Satisfaction with life", "Emotional support" and "Social support", which have enough correlation to measure the personal perception of quality of life associated with ICT use and are consistent with previous psychometric studies. The results of the TICO scale indicate that more than 70% of the sample feel ICT have united their family during home confinement and more than 45% experience happy feelings when they use ICT. In home confinement, ICT use has improved users' quality of life, mainly their satisfaction with life and social and family support.

## Introduction

The world is facing an unprecedented crisis in human history. Previous epidemics and pandemics, such as the 1918 influenza pandemic (also known as the Spanish flu), the HIV/AIDS pandemic of 1981, the SARS pandemic of 2002, the H1N1 influenza pandemic of 2010, the Ebola outbreak of 2014, among many others, do not measure up to the COVID-19 pandemic in terms of their impact on interpersonal relationships and, of course, on the use of ICT to mitigate their effects.

**Data Availability Statement:** The data for this manuscript are stored on: http://hdl.handle.net/10045/109886.

**Funding:** The authors received no specific funding for this work.

**Competing interests:** The authors have declared that no competing interests exist.

As Piña-Ferrer [1] points out, at this time of great global social uncertainty, the only vaccine could be fact-checked, truthful and scientific "information", but the reality is that the flow and, above all, the speed with which news and information is transmitted today exponentially raises fear of contagion and death. The perception of risk is directly proportional to the degree of exposure and to variables related to previous pathologies, age, and lifestyle.

According to Casero-Ripollés [2], regarding the flow information, the media disseminate a huge volume of news, many of them fake, that generate discontent and, above all, loss of trust and credibility among the population. The author concludes that the consequences of this dynamic are very negative for the democratic system, and, in our opinion, this situation also destabilizes people's quality of life.

Many countries quickly understood the need to implement responsible policies to try to prevent the spread of the virus through the simplest system: physical distancing. In this regard, it is important to note that "social distancing" has never really been the goal [3], since in most cultures it could be understood as a punishment and not as a measure of protection, given that ICT allow people to maintain social relations without physical contact.

So far ICT have become an active part in people's lives, albeit with a series of problems associated with them, such as possible addiction, nomophobia, techno-interference, the phantom phone vibration syndrome, and the fear of missing out (FOMO), among others [4–6].

Research shows us that the use of ICT significantly increases life satisfaction in the elderly population, generating important feelings of self-sufficiency [7]. Something similar happens with young people and adults, who use ICT as a tool of social and emotional support and as a source of social interactions, which improves their psychological well-being [8–10].

In new situations, new action systems are generated. ICT have made a giant leap in this crisis. Many of the actions that have been enhanced, in addition to telecommuting, are psychological crisis interventions and social support therapies through ICT, as reported by Inchausti, Macbeth, Hasson-Ohayon, and Dimaggio [11], as well as their new application by people who had not used them before [12].

In the face of the COVID-19 pandemic, it is necessary to distinguish between two major types of problems: to keep the population healthy against the virus and to maintain people's quality of life. We know that the concept of quality of life is broad, controversial and varies depending on the context and the perception of each person at a given time. Therefore, it is dynamic, subjective, and adaptable.

According to Schalock and Verdugo [13], there are more than 200 definitions of this construct, but multiple intercultural studies agree on its main dimensions: interpersonal relationships, psychological well-being, material well-being, physical well-being, personal development, social acceptance, self-determination and rights. All these dimensions are grouped by the authors into three factors: Independence (personal development and self-determination), social participation (relationships, social acceptance and rights), and well-being (psychological, physical and material).

For Brooks et al. [14], home confinement creates a series of problems that significantly affect people's quality of life, including the following:

- Fear of contagion and death, for them and their loved ones.

- Frustration and boredom, which increases without access to social media.

- Perception of information provided by health care officials as deficient.

- Stress and anxiety caused by shortage of protective supplies.

- Impaired mental health associated with the duration of home confinement.

Another interesting study carried out in Spain, Italy and the United Kingdom [15] high-lights the impact that home confinement can have on some relevant aspects of quality of life:

- More than 40% of the studied population are at risk of developing mental health problems.

- More than 65% considers that the information provided by public health officials in the different countries is deficient, and that what the population needs the most to improve their quality of life is to know specific plans to return to normality.

- More than 60% notes that even in this pandemic, health is no more important than the economy, which indicates that the economic factor is fundamental in quality of life.

Considering these observations, it is pertinent to ask whether the use of ICT during home confinement can change users' perception of quality of life. Therefore, the main objective of this work is to adapt and validate a psychometric scale that examines the influence of ICT use on quality of life during home confinement. According to its initials in Spanish, TICO is the acronym used to refer to this scale from now on.

The initial hypothesis is that ICT use improves the quality of life of the general population in a state of home confinement. To this end, we will study the influence of ICT on users' satisfaction with life in general, with active social participation, with subjective social support, and the extent to what ICT use can reduce users' sense of personal and social isolation, improve family dynamics and increase the psychological well-being of the confined population.

To measure Quality of Life, we consider three variables to be of the utmost importance in the state of confinement: loneliness and social support, psychological well-being and satisfaction with life.

Loneliness is associated with psychological and physical discomfort. During confinement many people have been alone and the only social link was through technology. We used a Scale of Solitude validated in adult population that studies family loneliness, marital loneliness and social loneliness [16], from which those items more representative of the state of loneliness and social support were adapted to the use of technologies.

We also adapted some items from the Ryff Psychological Welfare Scale, validated in Spanish populations [17]. The adaptation of the items was taken into account according to the fundamental variables of psychological well-being: self-acceptance, positive relationships, autonomy, control of the environment, purpose in life and personal growth, all in relation to the use of technologies.

Finally, from the Life Satisfaction Scale [18, 19] of the original for world population and the Spanish adaptation of the scale, items were extracted which were adapted to the situation of confinement and the use of technologies. The scale measures subjective psychological well-being, so it fits perfectly into our study for the confinement situation.

## Methods

### Procedure

The first step was to select items from the original scales developed by Cardona, Villamil, Henao and Quintero [16], Díaz et al. [17], Atienza, Balaguer and García-Merita [18], and Diener, Emmons, Larsen and Griffin [19], for their linguistic and cultural adaptation to the Spanish social and health context derived of the home confinement implemented in response to the COVID-19 pandemic, and to ICT use at home.

Three experts examined the validity of the scale's content. The instrument, objectives and characteristics of the research were presented to the selected experts, who studied and selected items based on their clarity, importance and relevance. Following the expert validation, 14

items were selected to be answered on a 7-point Likert scale, where 1 was "totally disagree" and 7 "totally agree". Items were presented as statements towards which respondents had to indicate their degree of agreement or disagreement. Finally, the 14-item scale was pilot tested among a group of 5 men and 5 women to ensure all statements were understandable. All participants stated the scale was significant to them.

## Sample and data collection

An exploratory and cross-sectional study has been carried out on a non-probabilistic and sequential sample (snowball). In this sampling method, which is used in cases when it is difficult to access or locate the population, data collection ends when the desired quotas are reached [20].

The process to obtain the sample was as follows: special care was taken to identify the groups, organizations and people, with whom to maintain initial contact and who could provide access to a sample that was as heterogeneous as possible according to the characteristics of the study (geographical scope Spain and all age groups from 18 years onwards). They were contacted and asked to participate in sharing and assisting in the dissemination of the survey. The web link to the survey was disseminated through

- Social networks: Twitter, Facebook and LinkedIn.

- By email.

- Through WhatsApp and Telegram groups

- An advertising campaign was carried out on Facebook, through the Business Manager platform, segmenting the population by age and geographical area.

Data was collected in Spain during the home confinement implemented in response to the COVID-19 pandemic from March 29 to May 10, 2020, using Google Forms. Consent was obtained from the University Miguel Hernández of Elche Project Evaluation Board, Registration 2020.228.E.OIR—Reference DPS.JGR.01.20.

In order to improve the response rate, invitations were sent out weekly following the ethical principles and code of conduct of the American Psychological Association [21], the first page of the online questionnaire informed participants of the objectives and importance of the research project, the estimated completion time (between 5 and 6 minutes), and the questions to answer. They were informed of their right to decline to participate and leave the online form at any time, also about the confidentiality and anonymization of the data. To participate and begin completing the form, respondents had to first state their agreement to do so.

The final sample obtained from the Spanish territory is 2,346 people, of whom 51.8% are women and 48.2% are men. The average age of the sample is 45 years (SD = 13.4), distributed as follows: 25% are 18 to 34 years old, 40% are 36 to 49 years old, 30% is 50 to 64 years old, and 5% are over 65.

More than half of respondents are married or have a common law partner (59.3%), 29.3% are single, 10.1% are divorced and 1.3% are widowed. The average number of people living with them during the home confinement is 3 (SD = 1.28). 66% of the sample are working outside or at home, while 17.7% are unemployed (in working age), 10.2% are students and 6.1% are retired. In terms of monthly income level, 37.5% earn from 1,000 to 2,000 euros, 30.9% earn more than 2,000 euros, 16.1% earn less than 1,000 euros, 9% have no income, and 6.4% did not answer.

97% of the sample remain in good health, with no symptoms of the coronavirus, 2.1% have mild symptoms, but have not been tested, and 0.5% claim to have had the disease but

experienced mild symptoms. In terms of education level, most respondents hold a bachelor's degree (53.6%), 23.4% hold a postgraduate or doctoral degree, 20.8% hold a secondary school or vocational training degree, while 2.8% only have basic education.

## Instrument

A purpose-created questionnaire has been designed for this study to collect information. It is divided into three blocks of information:

- Sociodemographic data.

- ICT use: type of technological devices used at home to connect to the Internet, media, social networks or messaging and video calling applications used during home confinement, frequency, and main reason for connection.

- Scale of quality of life and ICT use during home confinement (TICO) adapted from Cardona, Villamil, Henao and Quintero [16], Díaz et al. [17], Atienza, Balaguer and García-Merita [18] and Diener, Emmons, Larsen and Griffin [19].

## Data analysis

Data processing and analysis has been carried out with the Statistical Package for the Social Sciences (SPSS) and the Structural Equation Modeling Software (EQS). Internal consistency has been measured with Cronbach's alpha to determine the homogeneity of the scale items. The underlying factor structure study has been identified through exploratory factor analysis (EFA), followed by principal components analysis with varimax rotation. The Bartlett's sphericity test and Kaiser-Meyer-Olkin test have been calculated to determine whether the factorial model derived from the application of the scale is appropriate. The structure of the TICO scale has been analyzed with a confirmatory factor analysis using the following goodness of fit indices: the Satorra-Bentler's scaled chi-square test (SBχ2) [22], the Comparative fit index (R-CFI; values equal to or greater than .90 indicate an acceptable model) [23] the standardized root mean squared residual (SRMR; values less than .08 indicate an acceptable model fit), and the Root Mean Square Error of Approximation (R-RMSEA; values equal to or smaller than .06 indicate a perfect fit) [24].

## Results

### ICT use in home confinement

The most widely used ICT during the COVID-19 confinement have been television and the Internet. The devices used to connect to social networks and the Internet have been smartphones (96.4%), personal computers (82%), tablets (43.4%), smart TVs (35%) and video game consoles (11.3%). However, people do not get online in just one device, but in several devices that are used interchangeably throughout the day, with an average daily frequency ranging from 1 to 3 hours.

In terms of the use of social networks and instant messaging applications, there is an absolute dominance of WhatsApp among the population, with 99.2%, followed by YouTube (74%) and Facebook (67%). Instagram (48.4%) and Twitter (31.10%) were relegated to the last positions. Social media and messaging and video-calling apps have been mainly used to communicate with friends (83.8%) and family (82.3%), for work-related activities (71.2%), to watch movies or TV series (68.5%), to get the news (66.4%), to study (61.9%) and listen to music

(61.9%). Other recurring reasons for using social media are: reading (44.1%), sports or physical activities (39.9%), seek motivation or inspiration during the coronavirus crisis (36.4%), search for cooking recipes (29.3%), play games (19.9%), listen to podcasts (18.7%) and make online purchases (17.4%).

Despite the intense use of the Internet during the COVID-19 confinement, 71% of the participants stated that they have not devoted all their free time to the use of ICT. Only 29% admitted doing so. Importantly, 46.2% consider they have been happy when they have used ICT, and 72% consider that the use of ICT has brought their family members together during the COVID-19 pandemic.

## Quality of life and ICT use during the COVID-19 confinement. Differences by age and gender

The results of the survey based on the TICO scale (Table 3) show that the majority of respondents value their lives satisfactorily, in most respects, thanks to the use of ICT in home confinement (Mo = 5, Me = 5), and consider that ICT use improves the circumstances derived from the COVI-19 pandemic (Mo = 6, Me = 5). They feel that when they use ICT, they are more satisfied (Mo = 6, Me = 5) and achieve the things they consider important (Mo = 5, Me = 5). Most respondents believe that ICT help them fulfill their life goals during the COVID-19 pandemic (Mo = 5, Me = 5). If they had to live again in a situation of confinement they would continue to use ICT (Mo = 7, Me = 7), because they allow them to permanently communicate with other people (Mo = 7, Me = 6), to know that their personal network of contacts cares about them (Mo = 6, Me = 5), and to ask for help from family and/or friends (Mo = 7, Me = 6).

Boredom and loneliness connect with the use of ICT during home confinement as most respondents say they use ICT when they feel bored (Mo = 7, Me = 5) or lonely (Mo = 5, Me = 4). In fact, participants state that they manage to maintain an active social and family life thanks to the use of ICT, which allow them to set up meetings and celebrations with friends and family (Mo = 7, Me = 5).

Results are very heterogeneous regarding the use of ICT for the management of emotions, such as sadness or feeling loved, during confinement. Most respondents neither disagree nor agree with the statement "I use ICTs when I feel sad" (Mo = 4, Mo = 4), as shown in Table 3. The results are clearer regarding the use of ICT when respondents "do not feel loved", as most of them disagreement with this statement (Mo = 1, Me = 3).

An ANOVA (Table 1) was performed to determine whether the results obtained on the scale were related to the gender and age variables. The results of this analysis show that, in general terms, there are no differences. In all cases, the result of the independent scale of these two factors exceeds the significance level of 0.05. However, this does not apply to the items related to sadness, loneliness, feeling loved and social participation, which do differ according to age group and, particularly, gender.

## Internal consistency of the TICO scale

The reliability of the TICO scale has been assessed with Cronbach's alpha, which is a measure of internal consistency, in which a value of 0.889 (Table 2) is considered excellent [25, 26].

To determine the consistency of the scale more deeply, the Cronbach's alpha coefficient was recalculated by incorporating in the descriptive statistics the value it takes when each item is deleted independently. The objective is to determine whether the alpha goes up or down to improve its consistency. As Table 3 shows, the value of the coefficient goes down in all cases, except for the last two items, 13 and 14, so if any of them were removed, the overall consistency

**Table 1. ANOVA by gender and age.**

|  | F | Sig. |
|---|---|---|
| Whenever I feel sad, I use ICT. |  |  |
| Sex | 62.321 | .000 |
| Age | 11.645 | .000 |
| Whenever I feel lonely, I use ICT. |  |  |
| Sex | 26.538 | .000 |
| Age | 6.907 | .000 |
| Whenever I do not feel loved, I lean on ICT. |  |  |
| Sex | 28.493 | .000 |
| Age | 9.874 | .000 |
| I use ICT to settle on meetings, celebrations and parties with my friends and family. |  |  |
| Sex | 76.161 | .000 |
| Age | 50.728 | .000 |

Source: Authors' own creation.

**Table 2. Scale reliability.**

| Cronbach's Alpha | Cronbach's Alpha based on typified elements | N. of elements |
|---|---|---|
| .889 | .893 | 14 |

Source: Authors' own creation.

would be slightly higher. However, the consistency is already excellent as it is, so it was decided to keep these two statements in the scale. The same table shows the corrected homogeneity coefficient and the item-total correlation. In all cases the resulting values are not zero or negative, which means all items correlate with the total.

**Table 3. Descriptive statistics and item-total correlation of the TICO scale.**

|  | Min. | Max. | Mode | Median | Corrected item-total correlation | Cronbach's alpha if item is deleted |
|---|---|---|---|---|---|---|
| 1. In most respects, ICT make my life in confinement satisfactory. | 1 | 7 | 5 | 5 | .616 | .879 |
| 2. Life in confinement has improved thanks to ICT. | 1 | 7 | 6 | 5 | .603 | .880 |
| 3. I am more satisfied with my life in confinement when I use ICT. | 1 | 7 | 6 | 5 | .660 | .877 |
| 4. ICT help me get "important" things done in confinement. | 1 | 7 | 5 | 5 | .593 | .880 |
| 5. If I had to live in confinement again, I would continue using ICT. | 1 | 7 | 7 | 7 | .577 | .882 |
| 6. Thanks to ICT, I always have someone to talk to. | 1 | 7 | 7 | 6 | .600 | .880 |
| 7. Thanks to ICT, I feel people care about me. | 1 | 7 | 6 | 5 | .628 | .879 |
| 8. Thanks to ICT, I can ask for help from family and friends. | 1 | 7 | 7 | 6 | .613 | .880 |
| 9. Whenever I feel sad, I use ICT. | 1 | 7 | 4 | 4 | .632 | .878 |
| 10. Whenever I feel lonely, I use ICT. | 1 | 7 | 5 | 4 | .653 | .877 |
| 11. Whenever I do not feel loved. I lean on ICT. | 1 | 7 | 1 | 3 | .537 | .883 |
| 12. When I feel bored, I turn to ICT. | 1 | 7 | 7 | 5 | .595 | .880 |
| 13. ICT help me settle on meetings and celebrations with friends and family. | 1 | 7 | 7 | 5 | .403 | .890 |
| 14. ICT help me have a clear purpose and direction in life. | 1 | 7 | 5 | 5 | .334 | .892 |

Source: Authors' own creation.

**Table 4. KMO and Bartlett tests.**

| | |
|---|---|
| KMO measure of sampling adequacy | .903 |
| Bartlett's sphericity test | |
| Approximate Chi-square | 16357.819 |
| Degrees of freedom | 91 |
| Significance | .000 |

Source: Authors' own creation.

## Exploratory factor analysis

A factor analysis has been carried out to explore the validity of the construct quality of life and ICT use during confinement. The objective was to study its latent structure, identifying common factors and main components, and thus identify the different dimensions that could constitute and represent the concept appropriately.

The Kaiser-Meyer-Olkin (KMO) test and the Bartlett's sphericity test have been performed to evaluate whether the factorial model (or the extraction of factors), as a whole, is significant (see Table 4). The KMO index yielded a value greater than 0.9, which indicates the correlation between variables is strong and that the test is very good. Bartlett's test yielded a significance (p-value) of <0.05, so the model is significant and factor analysis can be applied.

For the selection of the number of components, we explored the eigenvalues obtained in Table 5 and the percentage of variance explained. As we can see, the first three components have variances (eigenvalues) greater than 1 so, following Kaiser's rule, they are the ones to be selected. In addition, these components or factors explain almost 64% of the variance of the original variables. The matrix of the unrotated factor structure (Table 6) indicates that all items have their highest weights in the first component, with values greater than .40 in all cases, which is the one that explains the most variance. Together with the Cronbach's alpha coefficient, this is a good indicator of the reliability and validity of the scale.

**Table 5. Total variance explained.**

| Component | Initial eigenvalues | | | Rotation sums of squared saturations | | |
|---|---|---|---|---|---|---|
| | Total | % of variance | Cumulative % | Total | % of variance | Cumulative % |
| 1 | 5.963 | 42.592 | 42.592 | 3.467 | 24.761 | 24.761 |
| 2 | 1.798 | 12.843 | 55.435 | 2.880 | 20.570 | 45.331 |
| 3 | 1.115 | 7.962 | 63.397 | 2.529 | 18.066 | 63.397 |
| 4 | .865 | 6.180 | 69.577 | | | |
| 5 | .783 | 5.593 | 75.170 | | | |
| 6 | .644 | 4.597 | 79.766 | | | |
| 7 | .483 | 3.453 | 83.220 | | | |
| 8 | .443 | 3.164 | 86.384 | | | |
| 9 | .395 | 2.821 | 89.205 | | | |
| 10 | .378 | 2.701 | 91.906 | | | |
| 11 | .354 | 2.530 | 94.436 | | | |
| 12 | .320 | 2.284 | 96.720 | | | |
| 13 | .293 | 2.092 | 98.812 | | | |
| 14 | .166 | 1.188 | 100.000 | | | |

Factor extraction method: Principal Components Analysis.

**Table 6. Matrix of components, without rotation and with Varimax rotation.**

| | Matrix of component (a) | | | Matrix of rotated components (a) | | |
|---|---|---|---|---|---|---|
| | Component | | | Component | | |
| Items | 1 | 2 | 3 | 1 | 2 | 3 |
| 1 | .701 | -.401 | | .769 | | |
| 2 | .689 | -.406 | | .805 | | |
| 3 | .736 | | -.325 | .805 | | |
| 4 | .676 | | | .730 | | |
| 5 | .660 | -.435 | | .721 | | .324 |
| 6 | .683 | | .475 | | | .768 |
| 7 | .703 | | .418 | | | .732 |
| 8 | .689 | | .486 | | | .781 |
| 9 | .684 | .572 | | | .865 | |
| 10 | .703 | .565 | | | .866 | |
| 11 | .593 | .575 | | | .852 | |
| 12 | .657 | | | | .580 | .352 |
| 13 | .472 | | .323 | | | .527 |
| 14 | .400 | | | .397 | | |
| | Extraction method: Analysis of main components. A 3 extracted components | | | Extraction method: Analysis of main components. Rotation method: Varimax with Kaiser normalization. A Rotation has converged on 5 iterations. | | |

Source: Authors' own creation.

The comparison, in the matrix of rotated components, of the relative saturations of each item in each of the factors shows that the first factor consists of items 1, 2, 3, 4, 5 and 14, which seem to reflect the dimension "Satisfaction with life". The second factor includes the variables 9, 10, 11 and 12, which could represent the "Emotional support" dimension. Finally, the third component consists of items 6, 7, 8 and 13, which tell us about the "Social Support" dimension.

## Confirmatory factor analysis

Based on the exploratory factor analysis, we propose a model structured around three correlated factors: Satisfaction with Life (6 items), Emotional Support (4 items) and Social Support (4 items). To confirm this model, a confirmatory factor analysis has been performed to compare it with an alternative one-dimensional model, in which all items are organized under a single factor. The goodness of fit (Table 7) has been assessed with the comparative fit index (R-CFI), the Satorra-Bentler's scaled Chi square index (SBχ2), the standardized root mean square residual (SRMR), and the Root mean square error of approximation (R-RMSEA). As

**Table 7. Goodness-of-fit indices for models.**

| Models | $\chi^2$ | $SB\chi^2$ | *df* | R-CFI | R-RMSEA (90% CI) | SRMR |
|---|---|---|---|---|---|---|
| 1 Factor | 5550.40 | 4316.13 | 77 | .664 | .153 (.143-.157) | .103 |
| 3 Correlated factors | 883.84 | 697.83 | 74 | .939 | .060 (.056-.064) | .050 |

Source: Authors' own creation.

**Table 8. Factor loadings for the three-factor model.**

| Items | Factor loadings |
|---|---|
| SV1 | .775 |
| SV2 | .790 |
| SV3 | .811 |
| SV4 | .712 |
| SV5 | .707 |
| BP6 | .364 |
| AP4 | .893 |
| AP5 | .918 |
| AP6 | .753 |
| AP7 | .624 |
| AP1 | .762 |
| AP2 | .788 |
| AP3 | .773 |
| PS1 | .441 |

All correlations are significant at the p < .01 level.

Source: Authors' own creation.

**Table 9. Factor correlations.**

| Factor | F1 | F2 |
|---|---|---|
| F1 | | |
| F2 | .467 | |
| F3 | .666 | .587 |

All correlations are significant at the p < .01 level.

Source: Authors' own creation.

shown in Tables 7–9, the model of three correlated factors has a good fit, within all the appropriate intervals, which does not occur with the one-factor model.

## Conclusions

This article contributes to the understanding of the influence of ICT on users' behaviors and personal perception of quality of life during such an exceptional and extreme situation as the home confinement implemented in response to the COVID-19 pandemic. Therefore, it offers a new approach to the quality of life construct in a previously unexplored area.

The main objective of this work was to analyze whether the TICO scale of Quality of Life and ICT use in confinement is a valid and reliable instrument to apply in the Spanish socio-cultural context. The results allow us to conclude that there is evidence of the psychometric quality of the scale. The estimated homogeneity analysis based on the item-total correlation coefficient confirmed its adequate levels in terms of the contribution of each item to the measurement of the construct. The evidence for the reliability of the TICO scale shows adequate consistency that facilitates its application.

On the other hand, the three factors examined in the instrument account for the following dimensions: "Satisfaction with life", "Emotional support" and "Social support". These dimensions present a sufficient correlation level to measure the personal perception of quality of life

associated with ICT use, as proposed in our theoretical model based on the exploratory factor analysis. This proposal is also consistent with the previous psychometric studies used as a starting point for the creation of this scale [16–19]. Therefore, we confirm the three-factor correlated model of our hypothesis.

We are aware that quality of life in relation to ICT use in confinement has increased over time in many areas. Some of the most significant are the elderly population [27–30], studies related to diet and nutrition [31, 32], physical activity [33] and education [34, 35] among others. Contrary to what could be thought a priori, in a state of confinement, ICT serve as a link between family members, as confirmed by 72% of the participants of our study. Likewise, almost 50% consider they are happy when they use ICT. Only 29% admit that they have invested all their free time in ICT consumption, which suggests that the majority of the population has been making a responsible use of ICT in this context of mandatory home confinement.

It is striking to note that the highest percentages of ICT use and consumption are related to social relations, which is what has been lacking the most in this situation, surpassing 80% of the cases studied. Participants perceive their quality of life is satisfactory thanks to the use of ICT, to the point that if they had to live a similar situation again, they would resort to ICT again. In general, people perceive that using ICT has helped them to improve the circumstances of their life in confinement, to meet many of their material needs, to stay in touch with friends and family, to realize other people care about their health, to feel accompanied and overcome boredom. Regardless of gender or age, the use of technologies during confinement has a positive influence on the assessment of quality of life. However, when we talk about emotions such as sadness, loneliness or the need to feel "loved", the use of technologies as a resource, support or personal relief shows a wide range of opinions that respond to age and gender differences.

Moreover, 97% of all participants stated they were in good health, with no COVID-19 symptoms. Therefore, as shown by previous studies [13, 14], physical well-being together with the social support perceived through ICT, and the satisfaction with life produced by the feeling of staying safe at home, influences the personal perception of a good quality of life during home confinement.

It can be concluded that ICT use in a state of confinement significantly improves the perception of quality of life, especially in terms of satisfaction with life and perceived emotional and social support.

The study is based on a large sample, representative of all Spanish regions and sociodemographic groups. However, it is not without limitations because sampling was not random. Thus, results should be considered as an initial approach with a large sample.

From the work carried out, the following future lines of research emerge: Study of the perception of Quality of Life in the post-Covid-19 /post confinement era; influence of ICTs on the modification of behavior and motivational levels; and perception of the time of use of ICTs and real use of them, differentiating the different social networks by use/consumption, taking into account age and gender.

## Supporting information

**S1 File.**
(DOC)

**S2 File.**
(DOC)

## Author Contributions

**Conceptualization:** José Antonio García del Castillo-Rodríguez, Irene Ramos-Soler, Carmen López-Sánchez, Carmen Quiles-Soler.

**Data curation:** José Antonio García del Castillo-Rodríguez, Irene Ramos-Soler, Carmen López-Sánchez, Carmen Quiles-Soler.

**Formal analysis:** José Antonio García del Castillo-Rodríguez, Irene Ramos-Soler, Carmen López-Sánchez, Carmen Quiles-Soler.

**Investigation:** José Antonio García del Castillo-Rodríguez, Irene Ramos-Soler, Carmen López-Sánchez, Carmen Quiles-Soler.

**Methodology:** José Antonio García del Castillo-Rodríguez, Irene Ramos-Soler, Carmen López-Sánchez, Carmen Quiles-Soler.

**Project administration:** José Antonio García del Castillo-Rodríguez, Irene Ramos-Soler, Carmen López-Sánchez, Carmen Quiles-Soler.

**Resources:** José Antonio García del Castillo-Rodríguez, Irene Ramos-Soler, Carmen López-Sánchez, Carmen Quiles-Soler.

**Software:** José Antonio García del Castillo-Rodríguez, Irene Ramos-Soler, Carmen López-Sánchez, Carmen Quiles-Soler.

**Supervision:** José Antonio García del Castillo-Rodríguez, Irene Ramos-Soler, Carmen López-Sánchez, Carmen Quiles-Soler.

**Validation:** José Antonio García del Castillo-Rodríguez, Irene Ramos-Soler, Carmen López-Sánchez, Carmen Quiles-Soler.

**Visualization:** José Antonio García del Castillo-Rodríguez, Irene Ramos-Soler, Carmen López-Sánchez, Carmen Quiles-Soler.

**Writing – original draft:** José Antonio García del Castillo-Rodríguez, Irene Ramos-Soler, Carmen López-Sánchez, Carmen Quiles-Soler.

**Writing – review & editing:** José Antonio García del Castillo-Rodríguez, Irene Ramos-Soler, Carmen López-Sánchez, Carmen Quiles-Soler.

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
