## [Decision Letter · Decision Letter 0]

30 Sep 2020

PONE-D-20-27286

Information and communication technologies and quality of life in home confinement: development and validation on the TICO scale

PLOS ONE

Dear Dr. Quiles-Soler,

Thank you for submitting your manuscript to PLOS ONE. After careful consideration, we feel that it has merit but does not fully meet PLOS ONE’s publication criteria as it currently stands. Therefore, we invite you to submit a revised version of the manuscript that addresses the points raised during the review process.

We look forward to receiving your revised manuscript.

Kind regards,

Frédéric Denis, Ph.D.

Academic Editor

PLOS ONE

Journal Requirements:

Reviewers' comments:

Reviewer's Responses to Questions

**Comments to the Author**

1. Is the manuscript technically sound, and do the data support the conclusions?

Reviewer #1: Yes

2. Has the statistical analysis been performed appropriately and rigorously? 

Reviewer #1: Yes

3. Have the authors made all data underlying the findings in their manuscript fully available?

Reviewer #1: Yes

4. Is the manuscript presented in an intelligible fashion and written in standard English?

Reviewer #1: Yes

5. Review Comments to the Author

Reviewer #1: Overall

This is a very timely manuscript. The manuscript is well written. The introduction is well developed.

Method

Provide a justification for selecting the particular scale for adaptation.

More detail is needed about recruitment of participants, i.e. how were the participants targeted – through organizations? Personal contacts? Etc.

What was the survey response rate?

Was the instrument administered anonymously or did you collect identifying information?

On average, how long did it take the participants to complete the instrument?

Conclusions

Given that the results indicated that age and gender were significant factors, comment on the differences in the conclusions.

Include recommendations for future research in this area.

6. PLOS authors have the option to publish the peer review history of their article (what does this mean?). If published, this will include your full peer review and any attached files.

Reviewer #1: No

---

## [Author Response · Author response to Decision Letter 0]

23 Oct 2020

Reviewer 1:

We have incorporated all of your suggestions into our revision. Thank you for your help.

---

## [Editor Report · Decision Letter 1]

26 Oct 2020

Information and communication technologies and quality of life in home confinement: development and validation on the TICO scale

PONE-D-20-27286R1

Dear Dr. Quiles-Soler,

We’re pleased to inform you that your manuscript has been judged scientifically suitable for publication and will be formally accepted for publication once it meets all outstanding technical requirements.

Kind regards,

Frédéric Denis, Ph.D.

Academic Editor

PLOS ONE
---

## [Editor Report · Acceptance letter]

28 Oct 2020

PONE-D-20-27286R1 

Information and communication technologies and quality of life in home confinement: development and validation of the TICO scale 

Dear Dr. Quiles-Soler:

I'm pleased to inform you that your manuscript has been deemed suitable for publication in PLOS ONE. Congratulations! Your manuscript is now with our production department. 

Kind regards, 

on behalf of

Dr. Frédéric Denis 

Academic Editor

PLOS ONE